# K-PLUG: Knowledge-injected Pre-trained Language Model for Natural Language Understanding and Generation

## Abstract

Existing pre-trained language models (PLMs) have demonstrated the effectiveness of self-supervised learning for a broad range of natural language processing (NLP) tasks. However, most of them are not explicitly aware of domain-specific knowledge, which is essential for downstream tasks in many domains, such as tasks in e-commerce scenarios. In this paper, we propose K-PLUG, a knowledge-injected pre-trained language model based on the encoder-decoder transformer that can be transferred to both natural language understanding and generation tasks. We verify our method in a diverse range of e-commerce scenarios that require domain-specific knowledge. Specifically, we propose five knowledge-aware self-supervised pre-training objectives to formulate the learning of domain-specific knowledge, including e-commerce domain-specific knowledge-bases, aspects of product entities, categories of product entities, and unique selling propositions of product entities. K-PLUG achieves new state-of-the-art results on a suite of domain-specific NLP tasks, including product knowledge base completion, abstractive product summarization, and multi-turn dialogue, significantly outperforms baselines across the board, which demonstrates that the proposed method effectively learns a diverse set of domain-specific knowledge for both language understanding and generation tasks. The code, data, and models will be publicly available[1].

## 1 Introduction

Pre-trained language models (PLMs), such as ELMo (Peters et al., 2018), GPT (Radford et al., 2018), BERT (Devlin et al., 2019), RoBERTa (Liu et al., 2019), and XLNet (Yang et al., 2019), have made remarkable breakthroughs in many natural language understanding (NLU) tasks, including text classification, reading comprehension, and natural language inference. These models are trained on large-scale text corpora with self-supervision based on either bi-directional or auto-regressive pre-training. Equally promising performances have been achieved in natural language generation (NLG) tasks, such as machine translation and text summarization, by MASS (Song et al., 2019), UniLM (Dong et al., 2019), BART (Lewis et al., 2020), T5 (Raffel et al., 2019), PEGASUS (Zhang et al., 2020), and ProphetNet (Yan et al., 2020). In contrast, these approaches adopt Transformer-based sequence-to-sequence models to jointly pre-train for both the encoder and the decoder.

While these PLMs can learn rich semantic patterns from raw text data and thereby enhance downstream NLP applications, many of them do not explicitly model domain-specific knowledge. As a result, they may not be as sufficient for capturing human-curated or domain-specific knowledge that is necessary for tasks in a certain domain, such as tasks in e-commerce scenarios. In order to overcome this limitation, several recent studies have proposed to enrich PLMs with explicit knowledge, including knowledge base (KB) (Zhang et al., 2019; Peters et al., 2019; Xiong et al., 2020; Wang et al., 2019; 2020), lexical relation (Lauscher et al., 2019; Wang et al., 2020), word sense (Levine et al., 2020), part-of-speech tag (Ke et al., 2019), and sentiment polarity (Ke et al., 2019; Tian et al., 2020). However, these methods only integrate domain-specific knowledge into the encoder, and the decoding process in many NLG tasks benefits little from these knowledge.

---

[1]Our code is available at `https://github.com/ICLR21Anonymous/knowledge_pretrain`.

To mitigate this problem, we propose a **K**nowledge-injected **P**re-trained **L**anguage model that is suitable for both Natural Language **U**nderstanding and **G**eneration (K-PLUG). Different from existing knowledge-injected PLMs, K-PLUG integrates knowledge into pre-training for both the encoder and the decoder, and thus K-PLUG can be adopted to both downstream knowledge-driven NLU and NLG tasks. We verify the performance of the proposed method in various e-commerce scenarios. In the proposed K-PLUG, we formulate the learning of four types of domain-specific knowledge: e-commerce domain-specific knowledge-bases, aspects of product entities, categories of product entities, and unique selling propositions (USPs) (Garrett, 1961) of product entities. Specifically, e-commerce KB stores standardized product attribute information, product aspects are features that play a crucial role in understanding product information, product categories are the backbones for constructing taxonomies for organization, and USPs are the essence of what differentiates a product from its competitors. K-PLUG learns these types of knowledge into a unified PLM, enhancing performances for various language understanding and generation tasks.

To effectively learn these four types of valuable domain-specific knowledge in K-PLUG, we proposed five new pre-training objectives: knowledge-aware masked language model (KMLM), knowledge-aware masked sequence-to-sequence (KMS2S), product entity aspect boundary detection (PEABD), product entity category classification (PECC), and product entity aspect summary generation (PEASG). Among these objectives, KMLM and KMS2S learn to predict the masked single and multiple tokens, respectively, that are associated with domain-specific knowledge rather than general information; PEABD determines the boundaries between descriptions of different product aspects given full product information; PECC identifies the product category that each product belongs to; and PEASG generates a summary for each individual product aspect based on the entire product description.

After pre-training K-PLUG, we fine-tune it on three domain-specific NLP tasks, namely, e-commerce knowledge base completion, abstractive product summarization, and multi-turn dialogue. The results show that K-PLUG significantly outperforms comparative models on all these tasks.

Our main contributions can be summarized as follows:

- We present K-PLUG that learns domain-specific knowledge for both the encoder and the decoder in a pre-training language model framework, which benefits both NLG and NLU tasks.

- We formulate the learning of four types of domain-specific knowledge in e-commerce scenarios: e-commerce domain-specific knowledge-bases, aspects of product entities, categories of product entities, and unique selling propositions of product entities, which provide critical information for many applications in the domain of e-commerce. Specifically, five self-supervised objectives are proposed to learn these four types of knowledge into a unified PLM.

- Our proposed model exhibits clear effectiveness in many downstream tasks in the e-commerce scenario, including e-commerce KB completion, abstractive product summarization, and multi-turn dialogue.

## 2 RELATED WORK

### 2.1 PLMS IN GENERAL

Unsupervised pre-training language model has been successfully applied to many NLP tasks. ELMo (Peters et al., 2018) learns the contextual representations based on a bidirectional LM. GPT (Radford et al., 2018) predicts tokens based on the context on the left-hand side. BERT (Devlin et al., 2019) adopts a bi-directional LM to predict the masked tokens. XLNet (Yang et al., 2019) predicts masked tokens in a permuted order through an autoregressive method. MASS (Song et al., 2019) pre-trains the sequence-to-sequence LM to recover a span of masked tokens. UniLM (Dong et al., 2019) combines bidirectional, unidirectional, and sequence-to-sequence LMs. T5 (Raffel et al., 2019) and BART (Lewis et al., 2020) present denoising sequence-to-sequence pre-training. PEGASUS (Zhang et al., 2020) pre-trains with gap-sentence generation objective. While human-curated or domain-specific knowledge is essential for downstream knowledge-driven tasks, these methods do not explicitly consider external knowledge like our proposed K-PLUG.

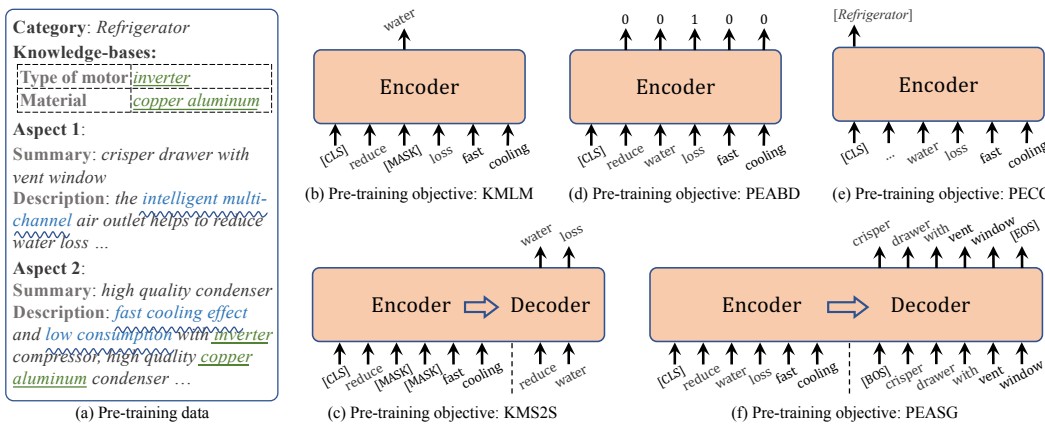

Figure 1: Pre-training data consists of 25 million textual product descriptions depicting multiple product aspects. We define knowledge as *e-commerce knowledge-bases*, *aspects of product entities*, *categories of product entities*, and *unique selling propositions of product entities*. Pre-training objectives include knowledge-aware masked language model (KMLM), knowledge-aware masked sequence-to-sequence (KMS2S), product entity aspect boundary detection (PEABD), product entity category classification (PECC), and product entity aspect summary generation (PEASG).

## 2.2 INJECTING KNOWLEDGE INTO PLMS

Recent work investigates how to incorporate knowledge into PLMs for NLU. ERNIE (Sun et al., 2019) enhances language representation with the entity/phrase-level masking. ERNIE (Zhang et al., 2019) identifies and links entity mentions in texts to their corresponding entities in KB. Similar to ERNIE (Zhang et al., 2019), KnowBERT (Peters et al., 2019) injects KBs into PLM. Xiong et al. (2020) leverages an entity replacement pre-training objective to learn better representations for entities. KEPLER (Wang et al., 2019) adopts the knowledge embedding objective in the pre-training. Besides, SKEP (Tian et al., 2020), SenseBERT (Levine et al., 2020), SentiLR (Ke et al., 2019), and K-ADAPTER (Wang et al., 2020) propose to integrate sentiment knowledge, word sense, sentiment polarity, and lexical relation into PLM, respectively. However, most of these studies are focused on integrating knowledge for language understanding task, work of utilizing domain-specific knowledge for pre-training for language generation tasks are limited. Inspired by these work, we construct K-PLUG that learns domain-specific knowledge into a PLM for both NLU and NLG tasks.

## 3 KNOWLEDGE-INJECTED PRE-TRAINING

In this section, we explain the data used to pre-train K-PLUG, its model architecture, and our pre-training objectives.

### 3.1 DATA PREPARATION

We collect the pre-training data from a mainstream Chinese e-commerce platform[2], which contains approximately 25 million textual product descriptions and covers 40 product categories. With an average length of 405 tokens, these product descriptions constitute a corpus with a size of 10B Chinese characters. Each product description consists of information on 10.7 product aspects on average, and each product aspect is accompanied with a summary highlighting its prominent features, as shown in Figure 1(a). Additionally, the e-commerce KB and USPs (further explained below) used in our pre-training data are as specified by the e-commerce platform and its online stores.

---

[2]https://www.jd.com/

## 3.2 MODEL ARCHITECTURE

K-PLUG adopts the standard sequence-to-sequence Transformer architecture (Vaswani et al., 2017), consisting of a 6-layer encoder and a 6-layer decoder as Song et al. (2019). We set the size of hidden vectors as 768, and the number of self-attention heads as 12. More details about the experimental settings are in the appendix.

## 3.3 KNOWLEDGE FORMULATION AND PRE-TRAINING OBJECTIVES

We formulate the learning of four types of knowledge in a unified PLM: e-commerce KB, aspects of product entities, categories of product entities, and USPs of product entities. Specifically, **e-commerce KB** stores standardized product attribute information, *e.g.*, (*Material*: *Cotton*) and (*Collar Type*: *Pointed Collar*). It provides details about the products (Logan IV et al., 2017). **Aspects of product entities** are features of a product, such as the *sound quality* of a stereo speaker, etc. (Li et al., 2020). **Categories of product entities** such as *Clothing* and *Food* are widely used by e-commerce platforms to organize their products so to present structured offerings to their customers (Luo et al., 2020; Dong et al., 2020) **USPs of product entities** are the essence of what differentiates a product from its competitors (Garrett, 1961). For example, a stereo speaker's USP exhibiting its supreme sound quality could be "*crystal clear stereo sound*". An effective USP immediately motivates the purchasing behavior of potential buyers.

We propose and evaluate five novel self-supervised pre-training objectives to learn the above-mentioned four types of knowledge in the K-PLUG model (see Figure 1).

**Knowledge-aware Masked Language Model (KMLM)**

Inspired by BERT (Devlin et al., 2019), we adopt the masked language model (MLM) to train the Transformer encoder as one of our pre-training objectives, which learns to predict the masked tokens in the source sequence (*e.g.*, "The company is [MASK] at the foot of a hill."). Similar to BERT, we mask 15% of all tokens in a text sequence; 80% of the masked tokens are replaced with the [MASK] token, 10% with a random token, and 10% left unchanged. Particularly, given an original text sequence $\boldsymbol{x} = (x_1, ..., x_m, ..., x_M)$ with $M$ tokens, a masked sequence is produced by masking $x_m$ through one of the three ways explained above, *e.g.*, replacing $x_m$ with [MASK] to create $\widetilde{\boldsymbol{x}} = (x_1, ..., [\text{MASK}], ..., \text{x}_M)$. MLM aims to model the conditional likelihood $P(x_m|\widetilde{\boldsymbol{x}})$, and the loss function is:

$$L_{MLM} = \log P(x_m|\widetilde{\boldsymbol{x}}) \tag{1}$$

The major difference from BERT is that our KMLM prioritizes knowledge tokens, which contain knowledge regarding product attributes and USPs, when selecting positions to mask and, in the case that the knowledge tokens make up less than 15% of all tokens, randomly picks non-knowledge tokens to complete the masking.

**Knowledge-aware Masked Sequence-to-Sequence (KMS2S)**

K-PLUG inherits the strong ability of language generation from the masked sequence-to-sequence (MS2S) objective. The encoder takes a sentence with a masked fragment (several consecutive tokens) as the input, and the decoder predicts this masked fragment conditioned on the encoder representations (*e.g.*, "The company [MASK] [MASK] [MASK] the foot of a hill.").

Given a text sequence $\boldsymbol{x} = (x_1, ..., x_u, ..., x_v, ..., x_M)$, a masked sequence $\widetilde{\boldsymbol{x}} = (x_1, ..., [\text{MASK}], ..., [\text{MASK}], ..., \text{x}_M)$ is produced by replacing the span $\boldsymbol{x}_{u:v}$, ranging from $x_u$ to $x_v$, with the [MASK] token. MS2S aims to model $P(\boldsymbol{x}_{u:v}|\widetilde{\boldsymbol{x}})$, which can be further factorized into a product $P(\boldsymbol{x}_{u:v}|\widetilde{\boldsymbol{x}}) = \prod_{t=u}^{v} P(x_t|\widetilde{\boldsymbol{x}})$ according to the chain rule. The loss function is:

$$L_{MS2S} = \sum_{t=u}^{v} \log P(x_t|\widetilde{\boldsymbol{x}}) \tag{2}$$

We set the length of the masked span as 30% of the length of the original text sequence. Similar to KMLM, KMS2S prioritizes the masking of text spans that cover knowledge tokens.

**Product Entity Aspect Boundary Detection (PEABD)**

A product description usually contains multiple product entity aspects. Existing work (Li et al., 2020) proves that product aspects influence the quality of product summaries from the views of importance, non-redundancy, and readability, which are not directly taken into account in language modeling. In order to train a model that understands product aspects, we leverage the PEABD objective to detect boundaries between the product entity aspects. It is essentially a sequence labeling task based on the representations of K-PLUG's top encoder layer.

Given a text sequence $x = (x_1, ..., x_M)$, the encoder of K-PLUG outputs a sequence $h = (h_1, ..., h_M)$, which is fed into a softmax layer, and generates a probability sequence $y$. The loss function is:

$$L_{PEABD} = -\sum_t \hat{y}_t \log y_t \tag{3}$$

where $y \in \{[0, 1]\}$ are the ground-truth labels for the aspect boundary detection task.

**Product Entity Category Classification (PECC)**

Product entity categories are the backbones for constructing taxonomies (Luo et al., 2020; Dong et al., 2020). Each product description document corresponds to one of the 40 categories included in our corpus, such as *Clothing*, *Bags*, *Home Appliances*, *Shoes*, *Foods*, etc. Identifying the product entity categories accurately is the prerequisite for creating an output that is consistent with the input.

Given a text sequence $x = (x_1, ..., x_M)$, a softmax layer outputs the classification score, $y$, based on the representation of the encoder classification token, [CLS]. The loss function maximizes the model's probability of outputting the true product entity category as follows:

$$L_{PECC} = -\hat{y} \log y \tag{4}$$

where $\hat{y}$ is the ground-truth product category.

**Product Entity Aspect Summary Generation (PEASG)**

Inspired by PEGASUS (Zhang et al., 2020), which proves that using a pre-training objective that more closely resembles the downstream task leads to better and faster fine-tuning performance, we propose a PEASG objective to generate a summary from the description of a product entity aspect. Unlike extracted gap-sentences generation in PEGASUS, our method constructs a more realistic summary generation task because the aspect summary naturally exists in our pre-training data.

Given an aspect description sequence $x = (x_1, ..., x_M)$, and an aspect summary sequence $y = (y_1, ..., y_T)$, PEASG aims to model the conditional likelihood $P(y|x)$. The loss function is:

$$L_{PEASG} = \sum_t \log P(y_t | x, y_{<t}) \tag{5}$$

## 4 EXPERIMENTS AND RESULTS

### 4.1 PRE-TRAINED MODEL VARIANTS

To evaluate the effectiveness of pre-training with domain-specific data and with domain-specific knowledge separately, we implement pre-training experiments with two model variants: C-PLUG and E-PLUG, whose configurations are the same as that of K-PLUG.

- **C-PLUG** is a pre-trained language model with the original objectives of MLM and MS2S, trained with a general pre-training corpus, CLUE (Xu et al., 2020), which contains 30GB of raw text with around 8B Chinese words.
- **E-PLUG** is a pre-trained language model with the original objectives of MLM and MS2S, trained with our collected e-commerce domain-specific corpus.

### 4.2 DOWNSTREAM TASKS

We fine-tune K-PLUG on three downstream tasks: e-commerce KB completion, abstractive product summarization, and multi-turn dialogue. The e-commerce KB completion task involves the

prediction of product attributes and values given product information. The abstractive product summarization task requires the model to generate a product summary from textual product description. The multi-turn dialogue task aims to output the response by utilizing a multi-turn dialogue context. The domain-specific knowledge we defined in this paper is essential for these tasks.

### 4.2.1 E-COMMERCE KB COMPLETION

**Task Definition.** E-commerce KB provides abundant product information that is in the form of (*product entity*, *product attribute*, *attribute value*), such as (*pid#133443*, *Material*, *Copper Aluminum*). For the E-commerce KB completion task, the input is a textual product description for a given product, and the output is the product attribute values.

**Dataset.** We conduct experiments on the dataset of MEPAVE (Zhu et al., 2020). This dataset is collected from a major Chinese e-commerce platform, which consists of 87,194 instances annotated with the position of attribute values mentioned in the product descriptions. There are totally 26 types of product attributes such as *Material*, *Collar Type*, *Color*, etc. The training, validation, and testing sets contain 71,194/8,000/8,000 instances, respectively.

**Model.** We consider the e-commerce KB completion task as a sequence labeling task that tags the input word sequence $x = (x_1, ..., x_N)$ with the label sequence $y = (y_1, ..., y_N)$ in the BIO format. For example, for the input sentence "*A bright yellow collar*", the corresponding labels for "*bright*" and "*yellow*" are *Color-B* and *Color-I*, respectively, and *O* for the other tokens. For an input sequence, K-PLUG outputs an encoding representation sequence, and a linear classification layer with the softmax predicts the label for each input token based on the encoding representation.

**Baselines.**

- **ScalingUp** (Xu et al., 2019) adopts BiLSTM, CRF, and attention mechanism to extract attributes.
- **JAVE** (Zhu et al., 2020) is a joint attribute and value extraction model based on a pre-trained BERT.
- **M-JAVE** (Zhu et al., 2020) is a multimodal JAVE model, which additionally utilizes product image information.

**Result.** Table 1 shows the experimental results. We observe that our K-PLUG performs better than baselines. C-PLUG achieves significantly better performance than BERT, which indicates that MS2S can also benefit the NLU task. E-PLUG outperforms C-PLUG, showing that training with domain-specific corpus is helpful. K-PLUG further exhibits a 2.51% improvement compared with E-PLUG. In short, we can conclude that the improvement is due to both the domain-specific pre-training data and knowledge-injected pre-training objectives.

Table 1: Experimental results with the F1 score for the e-commerce KB completion task. The results in the first block are taken from Zhu et al. (2020).

| Model | P | R | F1 |
|---|---|---|---|
| LSTM | 79.68 | 86.43 | 82.92 |
| ScalingUp | 65.48 | 93.78 | 77.12 |
| BERT | 78.27 | 88.62 | 83.12 |
| JAVE | 80.27 | 89.82 | 84.78 |
| M-JAVE | 83.49 | 90.94 | 87.17 |
| C-PLUG | 89.79 | 96.47 | 93.02 |
| E-PLUG | 89.91 | 96.75 | 93.20 |
| K-PLUG | **93.58** | **97.92** | **95.97** |

### 4.2.2 ABSTRACTIVE PRODUCT SUMMARIZATION

**Task Definition.** Abstractive product summarization task aims to capture the most attractive information of a product that resonates with potential purchasers. The input for this task is a product description, and the output is a condensed product summary.

**Dataset.** We perform experiments on the dataset of Li et al. (2020), which contains 1.4 million instances collected from a major Chinese e-commerce platform, covering three categories of product: *Home Appliances*, *Clothing*, and *Cases & Bags*. Each instance in the dataset is a (product information, product summary) pair, and the product information contains an image, a title, and other product descriptions. In our work, we do not consider the visual information of products. Notice that the task of abstractive product summarization and product entity aspect summary generation (PEASG) are partly different. The abstractive product summarization task aims to generate a complete and cohesive product summary given a detailed product description. Given a product aspect description, PEASG aims to produce an aspect summary that basically consists of condensed USPs. In addition, for abstractive product summarization task, the average length of the product summaries is 79, while the lengths of the product aspect summaries are less than 10 in general.

**Model.** Abstractive product summarization task is an NLG task that takes the product description as the input and product summary as the output.

**Baselines.**

- **LexRank** (Erkan & Radev, 2004) is a graph-based extractive summarization method.
- **Seq2seq** (Bahdanau et al., 2015) is a standard seq2seq model with an attention mechanism.
- **Pointer-Generator (PG)** (See et al., 2017) is a seq2seq model with a copying mechanism.
- **Aspect MMPG** (Li et al., 2020) is the-state-of-the-art method for abstractive product summarization, taking both textual and visual product information as the input.

**Result.** Table 2 shows the experimental results, including ROUGE-1 (RG-1), ROUGE-2 (RG-2), and ROUGE-L (RG-L) F1 scores (Lin & Hovy, 2003). K-PLUG clearly performs better than other text-based methods. E-commerce knowledge plays a significant role in the abstractive product summarization task, and domain-specific pre-training data and knowledge-injected pre-training objectives both enhance the model. K-PLUG achieves comparable results with the multimodal model, Aspect MMPG. The work of Li et al. (2020) suggests that product images are essential for this task, and we will advance K-PLUG with multimodal information in the future.

Table 2: Experimental results with the ROUGE score for the abstractive product summarization task. The results in bold are the best performances among the models taking only texts as the input, and * denotes the model taking both product images and texts as the input. The results in the first and second blocks are taken from Li et al. (2020).

| Model | Home Applications | | | Clothing | | | Cases&Bags | | |
|---|---|---|---|---|---|---|---|---|---|
| | RG-1 | RG-2 | RG-L | RG-1 | RG-2 | RG-L | RG-1 | RG-2 | RG-L |
| LexRank | 24.06 | 10.01 | 18.19 | 26.87 | 9.01 | 17.76 | 27.09 | 9.87 | 18.03 |
| Seq2seq | 21.57 | 7.18 | 17.61 | 23.05 | 6.84 | 16.82 | 23.18 | 6.94 | 17.29 |
| MASS | 28.19 | 8.02 | 18.73 | 26.73 | 8.03 | 17.72 | 27.19 | 9.03 | 18.17 |
| PG | 31.11 | 10.93 | 21.11 | 29.11 | 9.24 | 19.92 | 31.31 | 10.27 | 21.79 |
| Aspect MMPG* | 34.36 | 12.52 | 22.35 | 31.93 | 11.09 | 21.54 | 33.78 | 12.51 | 22.43 |
| C-PLUG | 32.75 | 11.62 | 21.76 | 31.73 | 10.86 | 20.37 | 32.04 | 10.75 | 21.85 |
| E-PLUG | 33.11 | 12.07 | 22.01 | 32.61 | 11.03 | 20.98 | 32.37 | 11.14 | 21.98 |
| K-PLUG | **33.56** | **12.50** | **22.15** | **33.00** | **11.24** | **21.43** | **33.87** | **11.83** | **22.35** |

### 4.2.3 MULTI-TURN DIALOGUE

**Task Definition.** The multi-turn dialogue task aims to output a response based on the multi-turn dialogue context (Shum et al., 2018). The input for this task is the dialogue context consisting of previous question answering, and the output is the response to the last question.

**Dataset.** We conduct experiments on two datasets of **JDDC** (Chen et al., 2020) and **ECD** (Zhang et al., 2018). **JDDC** is collected from the conversations between users and customer service staffs from a popular e-commerce website in China and contains 289 different intents, which are the goals of a dialogue, such as updating addresses, inquiring prices, etc, from after-sales assistance. There are 1,024,196 multi-turn sessions and 20,451,337 utterances in total. The average number of turns for each session is 20, and the average tokens per utterance is about 7.4. After pre-processing, the training, validation, and testing sets include 1,522,859/5,000/5,000 (dialogue context, response) pairs, respectively. **ECD** is collected from another popular e-commerce website in China and covers over 5 types of conversations based on 20 commodities. Additionally, for each ground-truth response, negative responses are provided for discriminative learning. The training, validation, and testing sets include 1,000,000/10,000/10,000 (dialogue context, response) pairs, respectively.

**Model.** We test with two types of K-PLUG: retrieval-based K-PLUG on the ECD dataset and generative-based K-PLUG on the JDDC dataset. For the retrieval-based approach, we concatenate the dialogue context and use [SEP] token to separate context and response. The [CLS] representation is fed into the output layer for classification. The generative-based approach is a sequence-to-

sequence model, which is the same as the model adopted in the abstractive product summarization task.

**Baselines.** The baselines also include both the retrieval-based (BM25, CNN, BiLSTM, and BERT) and generative-based approaches. Other baselines are as follows.

- **SMN** (Wu et al., 2017) matches a response with each utterance in the context.

- **DUA** (Zhang et al., 2018) is a deep utterance aggregation model based on the fine-grained context representations.

- **DAM** (Zhou et al., 2018) matches a response with the context based using dependency information based on self-attention and cross-attention.

- **IoI** (Tao et al., 2019) is a deep matching model by stacking multiple interactions blocks between utterance and response.

- **MSN** (Yuan et al., 2019) selects relevant context and generates better context representations with the selected context.

**Result.** Table 3 and 4 show the experimental results on the JDDC and ECD datasets, respectively. We report ROUGE-L (RG-L) F1, BLEU, and recall at position $k$ in $n$ candidates ($R_n@k$). We can observe that, both on the retrieval-based and generative-based tasks, K-PLUG achieves new state-of-the-art results, and e-commerce knowledge presents consistent improvements. K-PLUG is evidently superior to BERT, possibly due to BERT's lack of domain-specific knowledge for pre-training with the general MLM objective.

We further perform a human evaluation on the JDDC dataset. We randomly choose 100 samples from the test set, and three experienced annotators are involved to determine whether K-PLUG outperforms E-PLUG with respect to (1) relevance between the response and the contexts and (2) readability of the response. The results are shown in Table 5. We can see that the percentage of "Win", which denotes that the results of K-PLUG is better than E-PLUG, is significantly larger than "Lose" (p-value < 0.01 for t-test). Kappa values (Fleiss, 1971) confirm the consistency for different annotators.

Table 3: Experimental results for the multi-turn conversation task on the JDDC dataset. The results in the first block are taken from Chen et al. (2020).

| Model | RG-L | BLEU |
|---|---|---|
| BM25 | 19.47 | 9.94 |
| BERT | 19.90 | 10.27 |
| Seq2Seq | 22.17 | 14.15 |
| PG | 23.62 | 14.27 |
| C-PLUG | 25.47 | 16.75 |
| E-PLUG | 25.93 | 17.12 |
| K-PLUG | **26.60** | **17.80** |

Table 4: Experimental results for the multi-turn conversation task on the ECD dataset. The results in the first block are taken from Zhang et al. (2018).

| Model | $R_{10}@1$ | $R_{10}@2$ | $R_{10}@5$ |
|---|---|---|---|
| CNN | 32.8 | 51.5 | 79.2 |
| BiLSTM | 35.5 | 52.5 | 82.5 |
| SMN | 45.3 | 65.4 | 88.6 |
| DUA | 50.1 | 70.0 | 92.1 |
| DAM | 52.6 | 72.7 | 93.3 |
| IoI-local | 56.3 | 76.8 | 95.0 |
| MSN | 60.6 | 77.0 | 93.7 |
| BERT | 54.3 | 73.4 | 94.3 |
| C-PLUG | 62.7 | 76.8 | 95.0 |
| E-PLUG | 65.8 | 80.1 | 95.6 |
| K-PLUG | **73.5** | **82.9** | **96.4** |

Table 5: Human evaluation results (%). "Win" denotes that the generated response of K-PLUG is better than E-PLUG.

| Relevance | | | | Readability | | | |
|---|---|---|---|---|---|---|---|
| Win | Lose | Tie | Kappa | Win | Lose | Tie | Kappa |
| 29.00 | 21.00 | 50.00 | 0.428 | 7.00 | 2.00 | 91.00 | 0.479 |

### 4.3 ABLATION STUDIES

To better understand our model, we perform ablation experiments to study the effects of different pre-training objectives.

**Result.** The ablation results are shown in Table 6. We can conclude that the lack of any pre-training objective hurts performance across all the tasks. KMS2S is the most effective objective for the abstractive product summarization and generative conversation tasks since this objective is highly close to the essence of NLG. Product-aspect-related objectives, *i.e.*, PEABD and PEASG, contribute much to the abstractive product summarization task, which proves that this task requires comprehensively understanding the product description from the view of product aspects, going beyond individual tokens.

Table 6: Experimental results for ablation studies.

| Model | KB Completion F1 | Abstractive Product Summarization | | | | | | Multi-Turn Conversation | |
| | | Home Applications | | Clothing | | Cases&Bags | | | |
| | | RG-1 | RG-2 | RG-1 | RG-2 | RG-1 | RG-2 | RG-L | BLEU |
|---|---|---|---|---|---|---|---|---|---|
| K-PLUG | **95.97** | **33.56** | **12.50** | **33.00** | **11.24** | **33.87** | **11.83** | **26.60** | **17.80** |
| -KMLM | 95.88 | 33.52 | 12.43 | 32.87 | 11.20 | 33.75 | 11.70 | 26.43 | 17.62 |
| -KMS2S | 95.76 | 33.13 | 12.14 | 32.12 | 10.97 | 33.74 | 11.43 | 25.82 | 16.97 |
| -PEABD | 95.89 | 33.26 | 12.30 | 32.96 | 11.14 | 33.69 | 11.17 | 26.07 | 17.58 |
| -PECC | 95.59 | 33.24 | 12.17 | 32.25 | 11.12 | 33.59 | 11.18 | 26.02 | 17.16 |
| -PEASG | 95.48 | 33.39 | 12.36 | 32.57 | 11.16 | 33.78 | 11.45 | 26.12 | 17.38 |

## 5 CONCLUSION

We present a knowledge-injected pre-trained model (K-PLUG) that is a powerful domain-specific language model trained on a large-scale e-commerce corpus designed to capture e-commerce knowledge, including e-commerce KB, product aspects, product categories, and USPs. The pre-training framework combines masked language model and masked seq2seq with novel objectives formulated as product aspect boundary detection, product aspect summary generation, and product category classification tasks. Our proposed model demonstrates strong performances on both natural language understanding and generation downstream tasks, including e-commerce KB completion, abstractive product summarization, and multi-turn dialogue.

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

# A  APPENDIX

## A.1  EXPERIMENTS SETTINGS FOR PRE-TRAINING

We adopt GELU activation (Hendrycks & Gimpel, 2016) as in GPT (Radford et al., 2018). We use Adam optimizer (Kingma & Ba, 2015) with a learning rate of 5e-4, $\beta_1 = 0.9$, $\beta_2 = 0.98$, L2 weight decay of 0.01, learning rate warm-up over the first 10,000 steps and linear decay of the learning rate. The dropout probability is 0.1. The maximum sequence length is set to 512 tokens. Pre-training was performed with 4 Telsa V100 GPUs. The pre-training is done within 10 epochs, which takes around 10 days, and the fine-tuning takes up to 1 day. We use the beam search with a beam size of 5 for inference for the NLG tasks. Other hyper-parameters can be found in our code.

## A.2 CASE STUDIES

We present some examples from the test set of each task, with comparisons of the ground-truth result and the outputs produced by the models of E-PLUG and K-PLUG.

Table 7: Case study for the e-commerce KB complete task. The *product attribute* and the corresponding *attribute value* is presented as [*attribute value*]$_{product\ attribute}$. The K-PLUG model accurately complete the e-commerce KB, while the E-PLUG model sometimes fails. The translation texts are given below the original examples.

| | |
|---|---|
| Ground-truth | ECCO 防滑简约筒[短靴]$_{靴筒高度}$
(ECCO's non-slip simple [ankle boots]$_{shaft\ height}$) |
| E-PLUG | ECCO 防滑简约筒短靴$_{鞋跟高度}$
(ECCO's non-slip simple [ankle boots]$_{heel\ height}$) |
| K-PLUG | ECCO 防滑简约筒[短靴]$_{靴筒高度}$
(ECCO's non-slip simple [ankle boots]$_{shaft\ height}$) |
| Ground-truth | a21 [运动风]$_{风格}$[撞色]$_{图案}$ 风衣
(A21's [sports]$_{style}$ windbreaker jacket with [contrasting color]$_{design}$ style) |
| E-PLUG | a21 [运动风]$_{风格}$ [撞色风]$_{风格}$ 衣
(A21's sports [windbreaker]$_{style}$ jacket with [contrasting color style]$_{style}$ |
| K-PLUG | a21 [运动风]$_{风格}$[撞色]$_{图案}$ 风衣
(A21's sports [windbreaker]$_{style}$ jacket with [contrasting color]$_{design}$ style) |
| Ground-truth | 配合[微弹]$_{弹性}$ 的[棉质]$_{材质}$ 面料手感柔软顺滑
(made from [low-strech]$_{elasticity}$ [cotton fabric]$_{material}$ for a silky smooth touch) |
| E-PLUG | 配合[微弹]$_{裤型}$ 的[棉质]$_{材质}$ 面料手感柔软顺滑
(made from [low-strech]$_{pants\ fit}$ [cotton fabric]$_{material}$ for a silky smooth touch) |
| K-PLUG | 配合[微弹]$_{弹性}$ 的[棉质]$_{材质}$ 面料手感柔软顺滑
(made from [low-strech]$_{elasticity}$ [cotton fabric]$_{material}$ for a silky smooth touch) |

Table 8: Case study for the abstractive product summarization task (*Home Applications* category). The K-PLUG model generates summaries describing more information about *e-commerce knowledge bases* and *unique selling propositions of product entities*.

| Input Description | 帅康大吸力智能触控欧式抽油烟机，不锈钢材质，品质大于所见，多重贴心，310pa大风压，17立方大吸力，不碰头设计，53db静音，动力强，拢烟面积大，只留美味，不留烟味… 
 (Sacon's European style kitchen range hood. Extra powerful ventilation. Intelligent touch control. Stainless steel. Superior quality with extra reliability. 310 Pa air pressure. 17 $m^3$ suction. Protection against head bumping. Limited noise of 53 dB. Highly powered. Captures large volume of fumes. Leaves no cooking smells ...) |
|---|---|
| Ground-truth | 想要清新的厨房环境，那就试试这款大吸力油烟机吧！*310pa*大风压与*17立方米* 的大吸力的完美搭配，能够瞬间吸附烹饪过程中产生的油烟，让您清新烹饪，不做"黄脸婆"！ 
 (This kitchen range hood with *extra suction and ventilation* is ideal for a clean and fresh kitchen. Perfectly combining *310pa* of air pressure and *17m³* of suction power, it filters out grease before you realize it. Never have to worry again about keeping your loved ones from cooking fumes. ) |
| E-PLUG | 这款帅康油烟机，采用优质的不锈钢材质，精工细作，打造出简约大气的外观，尽显出欧式的厨房魅力，具有强劲的大吸力，快速吸走烹饪产生的油烟，智能的触控面板设计，使操作更加方便。 
 (This kitchen range hood from Sacon is made of *stainless steel* with cutting edge technology. Its design adds contemporary style to your European style kitchen. It features *extra suction and ventilation* for immediate filtering of fumes and grease. Its intelligent touch control panel makes it extremely easy to use. ) |
| K-PLUG | 这款不锈钢油烟机拥有*17立方*大吸力，快速清除油烟，给您一个清新的厨房环境。*310pa*大风压，保持排烟畅通，防止油烟倒灌。低至*53分贝*的运行声音，让您告别厨房的嘈杂。 
 (This *stainless steel* kitchen range hood features *extra suction and ventilation* of *17 $m^3$* for immediate filtering of grease and fumes, leaving your kitchen fresh. Its *310 pa* air pressure ensures clear filtering without backflow. Its low working noise of *53 dB* keeps your kitchen nice and quiet.) |
| Input Description | 康宝8升四季控温型，经济适用，家用厨房卫生间淋浴洗澡强排式燃气热水器，厨房洗菜，冬天洗漱再也不冻手，实打实的配置，高纯度无氧铜水箱，LED数字大屏，封闭式燃烧系统，防患于未然，控温清晰可见，多重安全防护，无氧铜水箱，让每次沐浴都是全新享受... 
 (Canbo's affordable 8L thermal control natural gas residential kitchen and bathroom water heater. Provides warm water for doing dishes and cleaning up in winter. High-quality accessories. High concentration oxygen-free copper tank. LED digital screen. Closed boiling system for safety. Easy temperature control. Multiple security checks. Oxygen-free copper tank. Lets you enjoy your showers ... ) |
| Ground-truth | 采用高纯度无氧铜水箱，增大了加热面积，高效速热且温度持久稳定。封闭式燃烧技术，提升了热效率，节能又高效。拥有控温功能，搭配*LED数字大屏*，让你能够根据需求来随意调节水温。 
 (High concentration *oxygen-free* tank with added heating area boils water quickly with stability. *Closed boiling technique* increases efficiency and saves energy. Thermal control and *LED digital screen* for adjustable water temperature anytime. ) |
| E-PLUG | 这款燃气热水器容量大，可以满足一年四季的用水需求。它采用封闭式燃烧技术，出热水的速度很快，沐浴的时候不用着急的等待，而且它在低压状态下也能正常运行，老小区用户也适用。 
 (This gas water heater has a large volume and provides warm water year round. It adopts the *closed boiling technique* for faster warming up and shorter wait time in shower. It works in low pressure condition and is suitable for older neighborhoods. ) |
| K-PLUG | 高配*LED数字大屏*，控温清晰可见，温度随时掌控于心。封闭式燃烧技术，快速度燃烧，加热效率更高。高纯度无氧铜水箱，加热速度更快。 
 (Equipped with *LED digital screen* for clear thermal control and temperature adjustment. *Closed boiling technique* speeds up warming and increases efficiency. High concentration *oxygen-free copper* tank for faster warming up.) |

Table 9: Case study for the abstractive product summarization task (*Clothing* category). The K-PLUG model generates summaries describing more information about *e-commerce knowledge bases* and *unique selling propositions of product entities*.

| | |
|---|---|
| Input Description | 劲霸男士t恤，夏季新品，撞色时尚，舒适，花卉印花，短袖，黑色，商务休闲，圆领，夏季，时尚都市，短袖t恤，悬垂感和耐穿性好，时尚圆领，简约大气，多色可选，莫代尔棉针织面料，柔滑触感，清凉舒爽，花卉印花结合，珠片绣花，时尚大方... 
 (K-Boxing's men's t-Shirt. This summer's new arrival. Contrasting colors. Soft and comfortable. Flower print. Short-sleeve. Black. Business casual. Crewneck. Summer. Urban style. Short-sleeve t-shirt. Naturally draping with high durability. Stylish crewneck. Basic and causal. Multiple color chioces. Modal knit fabric. Soft and elastic. Loose fit. Flower print. Glitter embroidery. Stylish and casual ....) |
| Reference | 选用莫代尔棉针织面料，手感柔滑，上身清凉舒爽，悬垂感好，耐穿性强。胸前撞色花卉印花结合珠片绣花，时尚又大方。经典圆领设计，自然突显型男风范。 
 (Made from *modal knit fabric*. *Incredibly soft and elastic*. Features a *slightly looser fit*. Natural draping with high durability. Decorated with flower print and *glitter embroidery* for a stylish and casual look. Finished with a classy *crewneck*, *appealing to the modern sense of fashion*) |
| E-PLUG | 圆领的领口更好的将颈部曲线所勾勒而出。水洗磨白的颜色平添了几分帅气与随性。让你和夏季的风一起随行。让休闲舒适的田园感油然而生。 
 (The *crewneck* design helps in elongating the neck. The light wash color gives an attractive yet effortless look. It dresses you up in a country asethetic style, comfy and relaxed just like the summer breeze. ) |
| K-PLUG | 采用莫代尔棉针织面料，柔滑触感，清凉舒爽，花卉印花结合珠片绣花，时尚大方，简约圆领设计，轻松修饰颈部线条。 
 (Made from *modal knit fabric*. *Incredibly soft and stretchy*. *Slightly looser fit*. Flower print and *glitter embroidery*. Stylish and casual. The basic *crewneck* design easily helps in *elongating the neck*. ) |
| Input Description | 吉普，羽绒服，男，中长款，90绒，冬季外套，新品，连帽，加绒，加厚，保暖羽绒外套，黑色，白鸭绒，时尚都市，型男都这么穿，融合艺术细节，创造76年传奇，潮流趋势必备，温暖新升级... 
 (Jeep's men's down jacket. Mid-thigh length. Filled with 90% down. Winter jacket. This winter's new arrival. Hoodedd. The down fill provides extra warmth. Warm down jacket. Black. White duck down. Urban style. Built for a perfect look. Designed with artistic details. Creating a legend for 76 years. A must-have to keep up with current fashion trends. Keeps you warmer than ever ... ) |
| Reference | 采用聚酯纤维面料，手感柔软，轻盈且透气性较好，穿在身上干爽舒适。内部以白鸭绒进行填充，充绒量较高，柔软蓬松，更有加厚修身的版型设计，保暖效果较好，为您抵御户外严寒。 
 (Made from *polyester*. *Feel soft*, lightweight, and breathable. Keeps you dry and comfortable. Filled primarily with *white duck down*, *fluffy and light*. Features a *thick yet slim-fit* design. Keeps you warm in cold climates. ) |
| E-PLUG | 这款羽绒服采用中长款的版型设计，修饰你的身材线条，而且还不乏优雅稳重气质。连帽的加持增添青春学院风气息。衣上字母印花的点缀，俏皮又减龄。 
 (This down jacket features a *mid-thigh length*, keeping a stylish silhouette and giving you an *elegant and mature look*. The *hood* and letter print on the jacket make you look younger. ) |
| K-PLUG | 采用聚酯纤维面料制成，手感柔软，亲肤透气。内部以白鸭绒填充，蓬松度高，轻盈温暖，更有连帽设计，可以锁住人体的热量，为您抵御户外寒冷天气，带来舒适的穿着体验。 
 (Made from *polyester*. *Soft and breathable*. Filled primarily with *white duck down*, fluffy and light. *Lightweight but warm*. Features a *hooded design*. Locks in the heat and keeps out the cold. Comfortable to wear. ) |

Table 10: Case study for the abstractive product summarization task (*Bags&Cases* category). The K-PLUG model generates summaries describing more information about *e-commerce knowledge bases* and *unique selling propositions of product entities*.

| Input Description | 菲安妮，秋季新款，斜跨包，女，印花磁扣，小方包，时尚单肩包，精致长款肩带，匀整车线工艺，高档ykk拉链，手感丝滑柔软且不脱色，优选进口水珠纹pvc，logo印花与包身融为一体，手感柔软舒适，防水耐磨，皮质肩带轻便减压，长度可调节，单肩/斜挎更舒心，平整均匀的车缝线技术
(Fion. This fall's new arrival. Corss body bag for women. Flower print magnetic snap closure. Square-shaped. Stylish tote bag. Well made long bag straps. Finished with flat lock stiching. Zippers produced by YKK. Flat and smooth surface. Anti-fading coloring. Made from imported PVC coated fabric. Logo print fits nicely. Soft and comfortable. Water-proof and durable. Lightweight leather shoulder strap. Adjustable length. 2 ways to carry. Finished with flat and even flatlock stiching.) |
|---|---|
| Reference | 采用进口水珠纹*pvc*面料制作，手感柔软舒适，防水耐磨耐用。品牌定制拉链，平滑顺畅不卡链。长款肩带，长度可调节，皮质轻便减压，单肩斜挎随心。
(Made from imported *PVC* coated fabric. *Soft and comfortable*. *Water-proof and durable*. Specially made smoothly meshing zippers. *Long* shoulder strap with *adjustable length*, made with leather for *light weight and reduced pressure*. 2 ways to wear: *cross body* or *with the top handles*. ) |
| E-PLUG | 这款单肩包采用了优质的*pvc*材料制作，它表面具有细腻的纹理，而它的包身上还具有精美的印花装饰，整体看上去非常优雅，而其内部空间也很大，所以带着它外出非常的方便。
(This *tote bag* is made friom imported high-qiuality *PVC* coated fabric. It has delicate texture and flower print decorations, giving it an elegant look. It can hold many items, making it ideal for every day use.) |
| K-PLUG | 优选进口水珠纹*pvc*，logo印花与包身融为一体，手感柔软舒适，防水耐磨，皮质肩带轻便减压，长度可调节，单肩斜挎更舒心，平整均匀的车缝线技术，细节之处彰显品质。
(Made from imported *PVC* coated fabric. Logo print fits nicely. *Soft and comfortable*. *Water-proof and durable*. Leather shoulder strap for *light weight and reduced pressure* with *adjustable length*. 2 ways to wear: *cross body* or *with the top handles*. Flat and even flatlocking finish shows its high quality. ) |
| Input Description | 美旅铝框拉杆箱，男女行李箱，20英寸，登机箱，万向轮，艺术旅行，坚韧抗压，打破时空的边界，突破风格的局限，灵活流畅，双排滑轮，内外兼修干湿分离，防护升级，包角设计，三重防爆锁扣，美观更牢固，复古亦时尚，且具有良好的密封性
(AmericanTourister's aluminum frame suitcase. For men and women. 20 Inch. Carry-on luggage. Spinner wheels. Essential for the creative journey. Strong and durable. Breaks the limits of time and style. Stable and effortless steering thanks to the doubled wheel design. Includes dividers for easy organization of packed items. Extra stability with corner protection. Triple locks for added security. Stylish design without compromising on durability. Classic yet fashionable. Closes and seals nicely. ) |
| Reference | 包角设计，显得更加简洁利落，而且又能有效[防撞耐摔]。铝合金框架结构，坚实牢固，具有良好的密封性。三重防爆锁扣更加稳固，让你出门安心无忧。内外间修，干湿分离，合理收纳更加有序。
(Conrer protection gives and basic and simple look with *added stability and durability*. *Aluminum alloy* frame makes it strong and *close and seal nicely*. *Triple locks* for added stability and security. Includes *dividers* for easy and organized packing.) |
| E-PLUG | 这款拉杆箱选用干净的黑色调，适合多种场合携带。精选材质，具有细腻的纹理质感，经久耐用。内部大空间处理，可以放置更多物品。
(This suitcase in *black* is suitable for various occasions. It is made from high-quality materials with delicate texture and [increased durability]. This large suitcase is built to provide room for extra items.) |
| K-PLUG | 采用铝合金框架结构，坚韧抗压，具有良好的密封性，配备三重防爆锁扣，美观更牢固，搭载内置式八轮设计，灵活流畅，搭配*tsa*海关密码锁，有效保护您的物品安全。
(Made with *aluminum alloy* frame for added stability. *Closes and seals nicely*. *Triple-lock* design to provide extra security. Stable and effortless steering thanks to the 8-wheel design. *TAS-approved locks* protect your items and cause no damage during security check.) |

Table 11: Case study for the multi-turn dialogue task on the ECD dataset. The K-PLUG model produces more accurate responses for the questions related to e-commerce knowledge bases.

| Q1 | 是纸皮核桃吗
(Is it paper-peel walnut?) |
|---|---|
| A1 | 是的哦
(Yes, it is.) |
| Q2 | 发的是特级吗
(Is it of premium quality?) |
| A2 | 什么意思
(What do you mean?) |
| Q3 | 核桃不是有大小吗我要送人去买10斤人在吗
(Aren't these walnuts sized differently? I'm getting 5 kilos. Are you still here?) |
| E-PLUG | 这是纸皮核桃手捏就开哦很好剥的呢
(This is paper-peel walnut, which you can peel easily with bare hands.) |
| K-PLUG | 一般直径在3-5cm
(They mostly have a diameter between 3-5 cm.) |
| Q1 | 这两个有什么区别
(What's the difference between these two?) |
| A1 | 您好上面的纯棉的尺寸1520
(Hi, the cotton one above has a size of 1520.) |
| Q2 | 这三种就是大小不一样啊
(These three only differ in size?) |
| E-PLUG | 是的哦亲以上是3种干巾的区别哦用途都一样呢
(Sure. That's exactly the difference between these three wiping products. They are made for the same purpose.) |
| K-PLUG | 中间的无纺布的尺寸1820 下面的天然纤维尺寸2020 加大加厚款
(The middle one made of nonwoven fabric has a size of 1820, and the bottom one made of natural fiber, with a size of 2020, is extra large and thick.) |
| Q1 | 这款茶刀包邮吗。
(Do you offer free shipping for this tea knife?) |
| A1 | 包邮的哦。
(Yes we do.) |
| Q2 | 多少钱啊
(How much is it?) |
| E-PLUG | 主要是我们30 箱之下就是这个价格了呢
(It's because we have this set price for all orders below 30 boxes.) |
| K-PLUG | 单把茶刀48哦
(It's 48 yuan for a single tea knife.) |

