# OpenReview forum: "K-PLUG: KNOWLEDGE-INJECTED PRE-TRAINED LANGUAGE MODEL FOR NATURAL LANGUAGE UNDERSTANDING AND GENERATION"
_ICLR.cc/2021/Conference — Reject_

### Official Review · AnonReviewer2 · 2020-10-26
**A well-designed and useful PLM for e-commerce; less clear if applicable to other fields**

**Rating:** 6
**Confidence:** 4

**Review:**


#### Summary

This paper introduces K-PLUG, an in-domain transformer-style pretrained language model. The domain on focus is e-commerce.

To enforce the PLM to be domain-aware, the authors identify several key types of knowledge for e-commerce: knowledge-bases, product aspects, product categories and product unique selling propositions (USP). They design specific pretraining objectives based on each type of knowledge.

Experiments on several downstream tasks exhibited the usefulness of the proposed model.

#### Strength

- The idea of domain-specific PLM is likely to be useful in practice but not widely explored. Specifically, in the e-commerce field people often have heavy needs for automated text processing and also a large amount of data, making this line of work more useful in practice.
- The pretraining objectives are properly designed, as they capture domain knowledge and have supervisions from the large-scale pretraining dataset. The experiment results also justified the design.

#### Weakness

- The proposed method (objectives) is hard to transfer to other domains. More importantly, it requires highly-structured and large-scale in-domain data for pretraining, which might not be available in other domains.
- Some downstream tasks are too similar to pretraining tasks. For instance, Abstractive Product Summarization looks very similar to Product Entity Aspect Summary Generation (PEASG) in pretraining, given that the image modality is ignored. I understand that if pretraining tasks are similar to downstream tasks, the performance on downstream tasks will be better. However, such similarity limits the ability of the experiments on downstream tasks to show the usefulness of the pretrained model. In other words, when the tasks are too similar, it becomes less clear whether the performance gain comes from the pretraining method or merely a larger training set. Also, have you tried directly using the pretraining dataset in downstream tasks (it looks possible for E-commerce KB Completion and Abstractive Product Summarization)?

#### Questions & Suggestions

- In Figure 1a, in the pretraining data aspect descriptions, the knowledge-base values and USPs are highlighted. How are these labeled? Are they directly available from the platform?
- For the pretraining dataset and downstream task datasets, each of them is said to be collected from "a Chinese e-commerce platform". Are they from the same or different platforms? It might be more clear to explicitly provide the platform names.
- In the formulation of KMLM and KMS2S (page 4) you mentioned "knowledge tokens", what are "knowledge tokens"? Are they knowledge-base tokens and/or USP tokens?
- For multi-turn dialog, since it is highly user-oriented, it would be better to have human evaluations.

#### Typos

- Page 5, PEASG, Line 6: "a aspect"

---

> ### Author Response · Authors · 2020-11-19
> **Response to Reviewer2**
>
> #Q1. The proposed method (objectives) is hard to transfer to other domains. More importantly, it requires highly-structured and large-scale in-domain data for pretraining, which might not be available in other domains.
>
> #Response: It is easy to extend our approach to other domains and others types of knowledge. In our paper, we define four types of knowledge, including e-commerce knowledge-bases, aspects of product entities, categories of product entities, and unique selling propositions (USPs) of product entities. We can find similar knowledge and structure in Wikipedia. Each entity in Wikipedia has a corresponding category and a knowledge-base, and the content for each entity is arranged by different aspects. The entity phrases can be regarded as USPs. In addition to our knowledge definition, our proposed pre-training objectives, including knowledge-aware masked language model (KMLM), knowledge-aware masked sequence-to-sequence (KMS2S), product entity aspect boundary detection (PEABD), and product entity category classification (PECC) could also be directly extended to Wikipedia. For the product entity aspect summary generation objective, we can obtain summary-like text for each aspect with the approaches in Zhang et al. (2020).
> Existing work on pre-training language models with Wikipedia mostly take the content in Wikipedia as flatted text or only use entity information. Our work presents a new perspective to pre-training language model with structured knowledge in Wikipedia.
> (Reference: Zhang, Jingqing, Zhao, Yao, Saleh, Mohammad, and Liu, Peter J. PEGASUS: Pre-training with extracted gap-sentences for abstractive summarization. ICML 2020.)
>
> #Q2. Abstractive Product Summarization looks very similar to Product Entity Aspect Summary Generation (PEASG) in pretraining. It is less clear whether the performance gain comes from the pretraining method or merely a larger training set. Also, have you tried directly using the pretraining dataset in downstream tasks?
>
> #Response: Although Abstractive Product Summarization task looks very similar to Product Entity Aspect Summary Generation (PEASG) in pre-training, they are totally two different tasks. The Abstractive Product Summarization task aims to generate a product summary given a detailed product description. It requires the model to sum up the key points in the input text, and produce a complete and cohesive text. PEASG aims to generate an aspect summary for a given product aspect, which consists of more condensed and abstractive USPs beyond the input text. In addition, for Abstractive Product Summarization task, the average length of the product summaries is 79, while the lengths of the product aspect summaries are less than 10 in general. We have performed experiments with directly using the pre-training dataset in abstractive product summarization task, and the results (RG-1/RG-2/RG-L) are as follows:
>
> |Pre-training model | Fine-tuning data | Home Applications|Clothing|Cases&Bags|
> |:----:|:----:|:----:|:----:|:----:|
> |W/o pre-training|PSUM|24.08/4.13/15.06|27.09/6.97/16.92|33.31/10.46/22.12|
> |W/o pre-training|PSUM&ASUM|21.37/3.97/13.75|22.07/4.82/14.51|22.04/4.79/14.40|
> |K-PLUG|PSUM|33.56/12.50/22.15|33.00/11.24/21.43|33.87/11.83/22.35|
>
> PSUM and ASUM denote product summarization dataset and aspect summarization dataset (pre-training data), respectively. We can conclude that directly using the pre-training dataset in the downstream product summarization task brings about negative influence on the product summarization models and pre-training with PEASG objective really boosts the performance.
>
> #Q3. How are the knowledge-base values and USPs labeled? Are they directly available from the platform?
>
> #Response: They are provided by the retailers or platform managers and are directly available from the platform.
>
> #Q4. For the pretraining dataset and downstream task datasets, each of them is said to be collected from "a Chinese e-commerce platform". Are they from the same or different platforms?
>
> #Response: Our pre-training dataset is from jd.com. The datasets of KB completion and JDDC dialogue are also from jd.com, and the dataset for ECD dialogue is from taobao.com, as stated in the corresponding papers. We have provided the platform name in our revised version.
>
> #Q5. What are "knowledge tokens"? Are they knowledge-base tokens and/or USP tokens?
>
> #Response: Yes, they are tokens containing knowledge regarding product attributes and unique selling propositions (USPs). We have explained it in our revised version.
>
> #Q6. Human evaluations for multi-turn dialogue.
>
> #Response: We have reported human evaluations in our revised version.

---

> > ### Comment · AnonReviewer2 · 2020-11-22
> > **Response to rebuttal**
> >
> > Thanks for the detailed explanations and even extra experiment results!
> >
> > For Q1, it's indeed an interesting and reasonable way to transfer the proposed methods to Wikipedia. Just to clarify, to me Wikipedia is more like a dataset or resource instead of a "domain"; what I meant in the original comment were domains like education, medication, etc., which might not have such large-scale and highly-structured data. Nevertheless, your ideas about transferring still make sense to me. I believe the paper will be more fascinating if you can describe your whole framework on a higher level that covers these ideas, for example, renaming and redefining your training objectives in a more generic way, and also trying to include experiments in other domains, so that people can see more clearly how to apply your framework in another domain.
> >
> > For Q2, I agree with you that these tasks are different. It might be helpful if you add a few lines to the paper to clarify this.
> >
> > Since my overall ideas didn't change significantly, I will keep my score.

---

> > > ### Author Response · Authors · 2020-11-23
> > > **Response to Reviewer2**
> > >
> > > Thanks for your response.
> > >
> > > ##Q1. There might not be such large-scale and highly-structured data in domains like education, medication.
> > >
> > > #Response: We can find similar knowledge structure in the PubMed Central (PMC) dataset (Lee at al., 2019) that contains 13.5B words (the corpus of English Wikipedia contains 2.5B words) from biomedical articles. We can regard the article sections in PMC as the aspects in our work,  and the biomedical entity tokens/phrases as USPs.
> > >
> > > (Reference: Lee et al., BioBERT: a pre-trained biomedical language representation model for biomedical text mining. arXiv:1901.08746)
> > >
> > >
> > > ##Q2. It might be helpful to clarify that these tasks are different.
> > >
> > > #Response: We have revised our paper to clarify the differences for these tasks.

---

> > > > ### Comment · AnonReviewer2 · 2020-11-23
> > > > **Response**
> > > >
> > > > (For Q1) Yes, these ideas are also very interesting, but I guess the settings are already quite different from this work, so it might need actual experiments to see if they can work as well. Anyway, this looks like an interesting direction that is worth investigating.

---

### Official Review · AnonReviewer1 · 2020-10-28
**K-PLUG Review**

**Rating:** 5
**Confidence:** 4

**Review:**


Overview:

The authors introduce K-PLUG, a knowledge-injected language model with domain-specific knowledge for both NLU and NLG tasks. They evaluate their method in e-commerce scenarios. They use five pre-training objectives, including domain-specific knowledge-bases, entities aspects, entities categories, and entities selling propositions.


Reasons to accept:
* The authors present K-PLUG that pre-trained on Chinese e-commerce data, which contains approximately 25 million textual product descriptions.

* The authors evaluate three different downstream tasks, e-commerce KB completion, abstractive product summarization, and multi-turn dialogue, and show good results compared to its baselines.

Reasons to reject:
* The paper is not easy-to-follow with too many abbreviations.

* The main concern of this work is: the comparison to existing works is too weak. As mentioned in Section 2.2, there are many related works also working on injecting knowledge or retrieving knowledge to improve language model pretraining. For example, KEPLER, SKEP, SenseBERT, and K-ADAPTER. However, the authors do not compare their training objectives/strategies to theirs, it is very hard for us to make a conclusion that which is better. It is not an apple-to-apple comparison. The only baselines to compare within this paper is C-PLUG and E-PLUG.

* The dataset used to pre-train the model contains expensive human annotation, which is not scalable. For example, the category labels (used for Product Entity Category Classification) and the summary (used for Product Entity Aspect Summary Generation). In other words, this kind of makes the pre-training process like a "multi-task" learning approach with human annotations, which in my opinion is hard to collect more data.

* The ablation study results that are shown in Table 5 basically suggest that many of the used objective functions are not important and could be ignored as the difference with and without is really marginal (I am sure some of them are not significantly better and could be the same if the authors tune a bit the hyper-parameters).

---

> ### Author Response · Authors · 2020-11-19
> **Response to Reviewer1**
>
> #Q1. The comparison to existing works is too weak.
>
> #Response: We have added more comparisons in our revised version. Related works on injecting knowledge in pre-training language models, such as KEPLER, SKEP, SenseBERT, and K-ADAPTER, can only be adopted to natural language understanding tasks, while our model is designed for both natural language understanding and generation tasks. We report more results of recent work on each task.
>
> #Q2. The dataset used to pre-train the model contains expensive human annotation, which is not scalable.
>
> #Response: The data we used to pre-train the model is produced by retailers without requiring further human annotations. In the real world, these information, like product entities, usually exit in certain form and is easy to access, like there are billions of product entities in total on the e-commerce platform we collect the data from. This work proposed an effective approach to utilize these information for NLG tasks.
>
> #Q3. The ablation study results that are shown in Table 5 basically suggest that many of the used objective functions are not important.
>
> #Response: Actually, some pre-training objectives may be complementary, e.g., for the KB completion task. These pre-training objectives have significant impacts on the downstream tasks, especially for Abstractive Product Summarization tasks. Even more interesting is that the most widely used pre-training objective, e.g., masked language model, contributes the least compared with others, which suggests that NLG-based pre-training objectives is effective for NLU tasks, which is interesting and may suggest further exploration of NLG-based pre-training objectives may be promising for pre-training language modeling.

---

### Official Review · AnonReviewer4 · 2020-10-29
**An interesting work should be described better**

**Rating:** 4
**Confidence:** 3

**Review:**

The paper proposes K-PLUG, a procedure for pre-training an encoder-decoder transformer with domain knowledge.
A main claimed contribution that domain-specific knowledge is incorporated both in the encoder and the decoder (as opposed to focusing on the encoder side).
The case study is e-commerce, where the authors incorporate information from a knowledge base about products.

More concretely, the authors propose to apply multiple pre-training objectives adapted to the knowledge source:
1. Token masking - similar to BERT, where the masked tokens correspond to `"knowledge tokens" rather than randomly selected tokens.
2. Encoder-decoder masking - Again, it is proposed to mask segments of text that cover `"knowledge tokens".
3. Product Entity Aspect Boundary Detection: tag mentions of `product aspects' in the text.
4. Product Entity Category Classification: Associating a given text with the product category
5. Product Entity Aspect Summary Generation: adapting previous work, this objective generates a summary from the description of a `"product entity aspect".

The approach is evaluated on several language understanding and generation tasks that pertain to the product knowledge source, including product knowledge base completion, abstractive product summarization, and multi-turn dialogue.

PROS:
- Integrating knowledge in transformer-based model is a topic of interest
- The experimental evaluation is solid. The authors compare with previous works, and present sensible ablation results.

CONS:
- Some details of the approach are under-specified/formulized. The results may therefore not be reproducible. (see comments)
- The paper is not self-contained (see comments)
- It is not clear if and how the approach extends to other types of knowledge, outside of the e-commerce and products domain.

Detailed comments:

- The major difference from BERT is that our KMLM prioritizes knowledge tokens - what are the `"knowledge tokens"? is it based on lexical match? does it only include `"content" words (with no prepositions, for example)? is ambiguity accounted for? is there a preliminary step of entity linking?

- What is a product `"aspect" exactly? how are `"attributes" and `"features" different?

- The baseline method JAVE is missing a citation

- Table 1: why not report also precision and recall, in addition to F1? there seems to be enough space.

- Table 2: note that the bold-faced results are not the best for all columns.

- The evaluation tasks are very briefly described. Perhaps examples can be given? with some explanation of how the proposed approach helps?

- Some terms are under-specified: e.g., `"intents" in Sec. 4.2.3.

- 'For the retrieval-based approach, we concatenate the dialogue context and use [SEP] token to separate context and response.
The [CLS] representation is fed into the output layer for classification.' - what is the context?

---

> ### Author Response · Authors · 2020-11-19
> **Response to Reviewer4**
>
> #Q1. It is not clear if and how the approach extends to other types of knowledge, outside of the e-commerce and products domain.
>
> #Response: It is easy to extend our approach to other domains and others types of knowledge. In our paper, we define four types of knowledge, including e-commerce knowledge-bases, aspects of product entities, categories of product entities, and unique selling propositions (USPs) of product entities. We can find similar knowledge and structure in Wikipedia. Each entity in Wikipedia has a corresponding category and a knowledge-base, and the content for each entity is arranged by different aspects. The entity phrases can be regarded as USPs. In addition to our knowledge definition, our proposed pre-training objectives, including knowledge-aware masked language model (KMLM), knowledge-aware masked sequence-to-sequence (KMS2S), product entity aspect boundary detection (PEABD), and product entity category classification (PECC) could also be directly extended to Wikipedia. For the product entity aspect summary generation objective, we can obtain summary-like text for each aspect with the approaches in Zhang et al. (2020).
> Existing work on pre-training language models with Wikipedia mostly take the content in Wikipedia as flatted text or only use entity information. Our work presents a new perspective to pre-training language model with structured knowledge in Wikipedia.
> (Reference: Zhang, Jingqing, Zhao, Yao, Saleh, Mohammad, and Liu, Peter J. PEGASUS: Pre-training with extracted gap-sentences for abstractive summarization. ICML 2020.)
>
> #Q2. What are the "knowledge tokens"? is it based on lexical match? does it only include "content" words (with no prepositions, for example)? is ambiguity accounted for? is there a preliminary step of entity linking?
>
> #Response: Knowledge-tokens are tokens containing knowledge regarding product attributes and unique selling propositions (USPs). It is based on lexical match. Most of them are content words. The influence of ambiguous words is limited, because we also allow masking for non-knowledge tokens in the pre-training objectives of KMLM and KMS2S. In addition, ambiguity do not influence objectives of PEABD, PECC, and PEASG. We perform entity linking for product entities.
>
> #Q3. What is a product "aspect" exactly? how are "attributes" and "features" different?
>
> #Response: "Aspects" are features of a product, such as the sound quality of a stereo speaker. They are not pre-defined by the e-commerce platform and are presented in the form of a complete text. "Attributes" are standardized for a certain product category and are presented as (attribute, value) pairs. In most cases, "features" are broader explanations of "attributes", and "attributes" are more concise "features".
>
> #Q4. The baseline method JAVE is missing a citation.
>
> #Response: Thanks for pointing this out. We have added the citation in our revised version.
>
> #Q5. Table 1: why not report also precision and recall, in addition to F1?
>
> #Response: We have reported the results of precision and recall in our revised version.
>
> #Q6. Table 2: note that the bold-faced results are not the best for all columns.
>
> #Response: The bold-faced results are the best among methods that use text-only input. We also listed the results of Aspect MMPG, but it takes both images and texts as the input so the comparison to it is not totally fair. And even so, our proposed method achieves comparable results to Aspect MMPG, demonstrating the effectiveness of our proposed method. We have illustrated that in the title of Table 2.
>
> #Q7. The evaluation tasks are very briefly described. Perhaps examples can be given?
>
> #Response: The examples are given in the Supplementary Material. We have included them in the appendix after the references in our revised version.
>
> #Q8. Some terms are under-specified: e.g., "intents" in Sec. 4.2.3.
>
> #Response: The "intents" denote the goals of a dialogue, such as updating addresses, inquiring prices, etc. We have revised our paper regarding the under-specified terms.
>
> #Q9. What is the context for "dialogue context"?
>
> #Response: The context is the dialogue history, e.g., previous turns of questions and answers in a multi-turn dialogue session.

---

### Official Review · AnonReviewer3 · 2020-10-29
**Good practice on pretraining language models for specific domain**

**Rating:** 5
**Confidence:** 4

**Review:**

Summary: This paper proposes pretraining language model for e-commerce domain. Specifically, the authors design five pretraining objectives to incorporate various domain knowledge into the the models with an encoder-decoder architecture. When further finetuned on language understanding and generation tasks in the e-commerce domain, the proposed models named K-PLUG outperforms the existing baseline models including those pretrained on general domains. The paper is generally easy to follow. Designs of the pretraining objectives are reasonable and empirically effective. Experiments are solid and convincing.

Pros:
The paper demonstrates a good practice on how to pretrain a language model for a specific domain by injecting a variety of structured and unstructured domain knowledge. Experiments verify that the knowledge-aware pretraining is indeed effective when you want to boost the performances of the tasks in specific domain. The experiments are well designed and quite comprehensive.

Concerns:
1. The paper is generally lack of novelty and inspiration, however. Most of the pretraining objectives have been tackled more or less in previous work. It is not surprising that when pretraining on the large amount of domain-specific data, the performance on the downstream tasks will improve. It will be more inspiring if the authors can give some high-level principles. Is it possible to design a more general framework that can work well when shifting to a new domain? What are the principles of designing pretrianing objectives with regard to domain knowledge and is there any way to evaluate them without need of expensive pretraining and finetuning?
2. The presentation of paper could be improved, especially for section 3. For example, from the notation it is unclear which training objectives are defined on encoder and which are on both encoder and decoder. Also in figure 1, there seems no clear distinction between the encoder and the decoder.

Additional questions:
1. How the five objectives are weighted in the pretraining?
2. What are the “knowledge-tokens” and “non-knowledge” tokens in KMLM and KMS2S? Are the results sensitive to them?

---

> ### Author Response · Authors · 2020-11-19
> **Response to Reviewer3**
>
> #Q1. Most of the pre-training objectives have been tackled more or less in previous work.
>
> #Response: Thanks for the comments. We also proposed a set of new and effective objectives. Knowledge-aware masked language model (KMLM) and knowledge-aware masked sequence-to-sequence (KMS2S) are inspired by BERT and MASS. The remaining pre-training objectives, including product entity aspect boundary detection (PEABD), product entity category classification (PECC), and product entity aspect summary generation (PEASG), are innovative objectives, which are proposed to incorporate the knowledge of product entity aspect and product entity category into the pre-training model. The experimental results have proven the effectiveness of these pre-training objectives.
>
> #Q2. Is it possible to design a more general framework that can work well when shifting to a new domain? What are the principles of designing pre-training objectives with regard to domain knowledge and is there any way to evaluate them without need of expensive pretraining and finetuning?
>
> #Response: There are two principles of designing pre-training objectives. The first is to train the model to learn general language representation, like the masked language model and masked sequence-to-sequence objectives. The second principle is to incorporate domain-specific knowledge, like the proposed product entity aspect boundary (PEABD), product entity category classification (PECC), and product entity aspect summary generation (PEASG) pre-training objectives. It is possible to design a more general framework, as in this work, and generalization makes it easier for domain transferring. We also try to make it adequately equip the model with various kinds of domain-specific knowledge. As for evaluation, to reduce consumption of expensive pre-training and fine-tuning, we can evaluate the pre-training objectives with a small set of data or pre-train with early-stopping for fewer training iterations.
>
> #Q3. The presentation of paper could be improved.
>
> #Response: Thanks for your suggestion. The presentations of section 3 and figure 1 have been improved, especially for the notations related to the encoder and decoder.
>
> #Q4. How the five objectives are weighted in the pretraining?
>
> #Response: They are equally weighted.
>
> #Q5. What are the “knowledge-tokens” and “non-knowledge” tokens in KMLM and KMS2S? Are the results sensitive to them?
>
> #Response: Knowledge-tokens are tokens containing knowledge regarding product attributes and unique selling propositions (USPs). We conduct experiments with the E-PLUG model that is pre-trained with the original objectives of MLM and MS2S only (without prioritizing knowledge tokens when masking words), and the results show the effectiveness of KMLM and KMS2S, suggesting that prioritizing knowledge-tokens do make a difference.

---

### Author Response · Authors · 2020-11-19
**Overall Response: Summary of the Revisions**

We appreciate all the reviewers for their insightful reviews. Here we summarize the main revisions we have made in the revised version of our paper.
1.	We have improved the presentations of section 3 and figure 1, especially for the notations related to the encoder and decoder.
2.	We have specified some terms, including “knowledge-tokens”, “non-knowledge-tokens”, "intents", etc.
3.	We have added citation for the baseline method JAVE.
4.	We have reported the results of precision and recall scores in Table 1.
5.	We have included some examples in the appendix after the references.
6.	We have included more comparative methods for the downstream tasks.
7.	We have included human evaluations for multi-turn dialogue.

---

### Decision · Program_Chairs · 2021-01-07
**Final Decision**

**Decision:**

Reject

**Comment:**

This paper proposes a new method for pre-training of language models in the e-commerce domain. It introduces five objectives for pre-training by incorporating domain knowledge into the model.

Pros • The paper is generally easy to follow. • Design of the pre-training objectives is reasonable. • Experimental results are solid and convincing. • A useful method is proposed, and its effectiveness has been verified in the e-commence domain.

Cons • Novelty of the work might not be enough. • Presentations can be improved.
• It is not clear whether the proposed approach can be applied to other domains which may not have enough structured data.

The authors have made several things clearer in the rebuttal. They have also added new experimental results. However, the overall quality of the paper does not reach the level of ICLR from the viewpoint of novelty, significance, and clarity.